# Discovery of novel benzophenone integrated derivatives as anti-Alzheimer's agents targeting presenilin-1 and presenilin-2 inhibition: A computational approach

Reshma Mary Martiz[1,2], Shashank M. Patil[1], Ramith Ramu[1]*, Jayanthi M. K.[3], Ashwini P.[2], Lakshmi V. Ranganatha[4], Shaukath Ara Khanum[5], Ekaterina Silina[6], Victor Stupin[7], Raghu Ram Achar[8]

1 Department of Biotechnology and Bioinformatics, School of Life Sciences, JSS Academy of Higher Education and Research, Mysuru, Karnataka, India, 2 Department of Microbiology, School of Life Sciences, JSS Academy of Higher Education and Research, Mysuru, Karnataka, India, 3 Department of Pharmacology, JSS Medical College, JSS Academy of Higher Education and Research, Mysuru, Karnataka, India, 4 Department of Chemistry, The National Institute of Engineering, Mysuru, Karnataka, India, 5 Department of Chemistry, Yuvaraja's College (Autonomous), University of Mysore, Mysuru, Karnataka, India, 6 Department of Human Pathology, I.M. Sechenov First Moscow State Medical University (Sechenov University), Moscow, Russia, 7 Department of Hospital Surgery 1, N.I. Pirogov Russian National Research Medical University (RNRMU), Moscow, Russia, 8 Division of Biochemistry, School of Life Sciences, JSS Academy of Higher Education and Research, Mysuru, Karnataka, India

* ramithramu@gmail.com

**Data Availability Statement:** All relevant data are within the manuscript and its Supporting Information files.

## Abstract

The most commonly accepted hypothesis of Alzheimer's disease (AD) is the amyloid hypothesis caused due to formation of accumulation of Aβ42 isoform, which leads to neurodegeneration. In this regard, presenilin-1 (PSEN-1) and -2 (PSEN-2) proteins play a crucial role by altering the amyloid precursor protein (APP) metabolism, affecting γ-secretase protease secretion, finally leading to the increased levels of Aβ. In the absence of reported commercial pharmacotherapeutic agents targeting presenilins, we aim to propose benzophenone integrated derivatives (BIDs) as the potential inhibitors of presenilin proteins through *in silico* approach. The study evaluates the interaction of BIDs through molecular docking simulations, molecular dynamics simulations, and binding free energy calculations. This is the first ever computational approach to discover the potential inhibitors of presenilin proteins. It also comprises druglikeliness and pharmacotherapeutic potential analysis of the compounds. Out of all the screened BIDs, BID-16 was found to be the lead compound against both the presenilin proteins. Based on these results, one can evaluate BID-16 as an anti-Alzheimer's potential specifically targeting presenilin proteins in near future using *in vitro* and *in vivo* methods.

**Funding:** The author(s) reported there is no funding associated with the work featured in this article.

## Introduction

Alzheimer's disease (AD) is a progressive neurodegenerative disease, which results in the gradual disruption in neuron's structure and function ultimately leading to the death of the neuron. It is the most common type of dementia found in nearly 60–70% of the cases according to the WHO report. According to the Alzheimer's association report, mortality rate has been increased by 146.2% between 2000 to 2019, with nearly 44 million people globally suffering till date [1].

The pathological cause for Alzheimer is the formation of extracellular amyloid plaques (Aβ) and cytoplasmic neurofibrillary tangles (NFT'S). The protein fragments of beta-amyloid accumulates outside the neurons, whereas tau protein accumulates inside them [1, 2]. Based on the recent clinical evidence, it is known that the progression of Alzheimer's disease is due to the formation of beta-amyloid plaques, which are formed by the cleavage of amyloid precursor protein (APP) by β-secretase and γ-secretase protease enzymes [3]. The Aβ42 oligomer isoform is the major toxic constituent in neurodegeneration AD. The toxic Aβ oligomers interact with glial cells causing inflammatory cascades, oxidative stress due to increased production of reactive oxygen species (ROS) and reactive nitrogen species (RNS). This causes the loss of function in many antioxidant defence enzymes, dysfunction in calcium metabolism which together leads to neuron degradation. Furthermore, these events cause dysfunction in APP metabolism leading to Aβ neurotoxic peptides causing AD [4, 5].

From recent advances in research, it has become evident that PSEN-1 gene, which is located on chromosome 14q24.2 and encodes PSEN-1 protein subunit of γ-secretase, is the main cause for early-onset of AD [6]. After mapping of the PSEN-1, it was found that PSEN-2 with 60% homology to that of PSEN-1 was found on chromosome 1q42.13 encoding its γ-secretase subunit responsible for the development of AD [7]. The PSEN-1 plays a crucial role in the pathogenesis of AD as it is found with over 300 mutations [8]. Recently, it is reported that both PSEN-1 and PSEN-2 are the functional subunits of γ-secretase [9]. As per the study conducted by **Silveyra et al. (2008)** [10], the correlation between PSEN-1 and AChE indicates that PSEN-1 can physically interact with AChE with little understanding on its mode of interaction. According to **Belluti et al. (2014)** [5], benzophenones-based derivatives can be a new promising target for AD, which inhibit AChE at sub-micromolar level.

Benzophenones are a group of compounds with more than 300 biologically active members with a common phenol-carbonyl-phenol skeleton and excellent structural diversity. These derivatives obtained from natural and/or synthetic sources are biologically active molecules displaying antitumor, anti-inflammatory, and antidiabetic activities. Several analogues of benzophenones are under clinical study; Combretastatin A-4 is well-known to exhibit antiangiogenic effects and is being studied in clinical trials. Benzophenones associated with other heterocyclic analogues possess potential applications rather than benzophenone alone. In this context, to pursue a potential approach towards the discovery of most efficient pharmacologically active compounds, the benzophenones are amalgamated with a new class of heterocyclic compounds. The presence of indole, oxadiazole, and benzimidazole nucleus with benzophenone in numerous categories of therapeutic agents has made it an essential anchor for the development of new pharmacophore [11–14]. In this work, we investigate the interaction of the benzophenones derivatives by using *in silico* approach using computational tool to analyse and elucidate the biological property of benzophenones derivatives with the Presenilin (PSEN) proteins. So far, this is the first *in silico* investigation on the inhibition of PSEN proteins.

## Materials and methods

### Homology modelling and structure validation

The protein sequences of PSEN-1 (ID: P49768-1) and PSEN-2 (ID: P49810-1) were obtained from UniProt (https://www.uniprot.org/) for which template search was completed using BLAST tool to find the best template. The Human gamma-secretase in complex with small molecule avagacestat (PDB: 6LQG) which had sequence identity of 100% for PSEN-1 and 71.23% for PSEN-2 was selected. The sequence was aligned to the template protein and the models were built using SWISS-MODEL (https://swissmodel.expasy.org/) [15]. The structures were energy computed using GROMOS96 43B1 parameter using SWISS PDB Viewer [16]. Further, the modelled structures were validated based on Ramachandran plot (**Figs 1 and 2**) to get a better understanding of the structure compatibility.

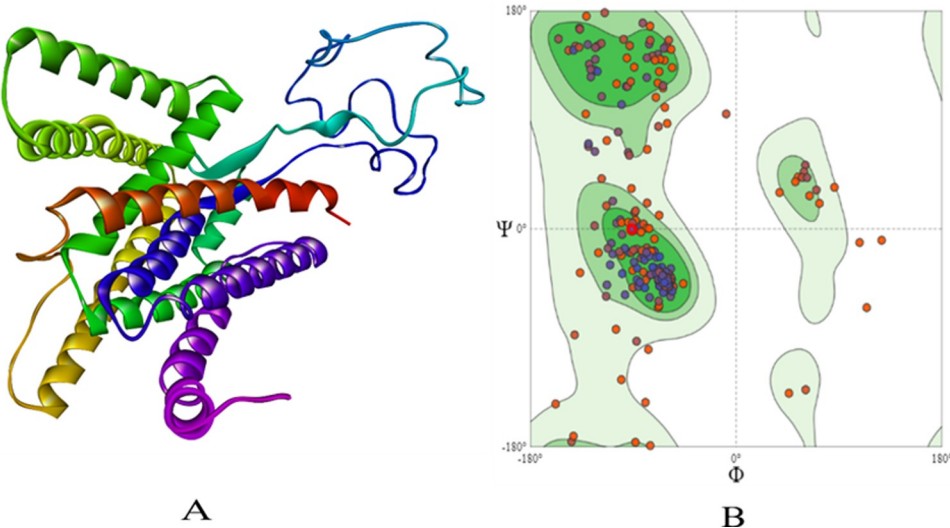

**Fig 1.** The homology built 3D structure of PSEN-1 is represented in (**A**). The structure assessment by Ramachandran plot is shown in (**B**).

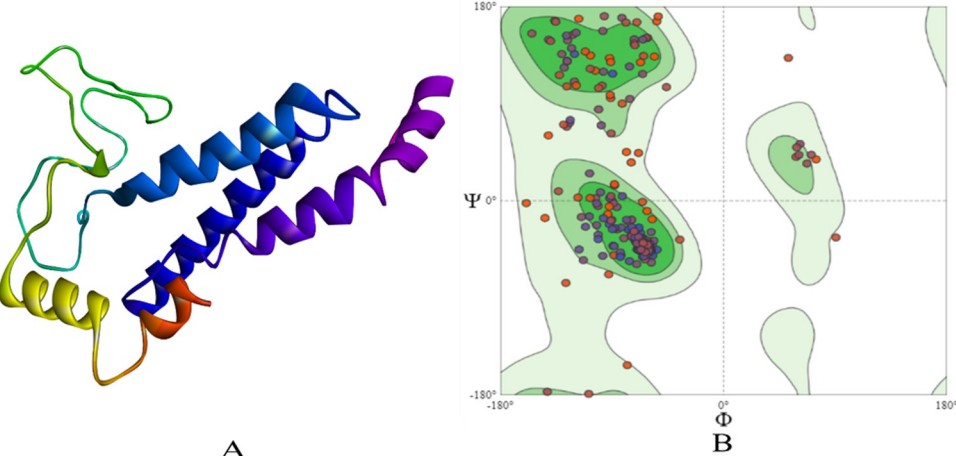

**Fig 2.** The homology built 3D structure of PSEN-2 is represented in (**A**). The structure assessment by Ramachandran plot is shown in (**B**).

### Prediction of binding site

To know the topographic feature and to get the better understanding of the binding pockets of the protein, Computed Atlas of Surface Topography of proteins (CASTp) (**http://sts.bioe.uic.edu/castp/index.html?3trg**) [17] server is used. The server identifies and measures the volume and area of all the available binding pockets. The program was used to get the binding site of PSEN-1 and PSEN-2 proteins.

### Molecular docking simulations

Molecular docking for the modelled PSEN-1 and PSEN-2 protein with the compounds was performed using AutoDock Vina 1.1.2 [18]**,** which uses Broyden-Fletcher-Goldfarb-Shanno algorithm to get a better insight of the conformational changes and to understand their interaction with the compound within the specified active site region. The pre-docking preparation for both proteins and ligands was done according to Autodock 4.2 [19] protocol using Autodock Tools 1.5.6, by removing all the water molecules and hetero atoms present in the proteins. Further, non-polar hydrogens and carbon atoms were merged to obtain the stability. The Kollmann united and Gasteiger-Marsili empirical atomic partial charges that were predicted by the Autodock Tools 1.5.6 were assigned to balance the structure. Similarly, the torsions were kept as default as given by the program, AD4 atom type were added to determine the types of atoms in the macromolecule, and the prepared structures were obtained in PDBQT format with inclusion of charges Q and AutoDock 4 atom types T for molecular docking simulation.

Based on the CASTp server prediction, the grid box for the binding pockets was prepared. For PSEN-1, the grid box of size x = 38.16 Å, y = 38.16 Å, z = 38.16 Å were set centred to cover the binding pockets and all essential residues at x = 174.73 Å, y = 180.58 Å, and z = 144.29Å. For PSEN-2 grid box of size x = 26.43 Å, y = 26.43 Å, z = 26.43 Å were set centred to cover the binding pockets and all essential residues at x = 162.88 Å, y = 178.42 Å, z = 152.74 Å. For the docking process, the ligand was considered to be flexible while protein was kept rigid. The binding analysis for PSEN-1 and PSEN-2 was performed using Biovia Discovery Studio Visualizer 2021 for the best docked conformation based on their binding affinity, total number of non-bonding interactions, and their hydrogen bonds.

### Molecular dynamics simulation

Molecular dynamics (MD) simulations were performed for the predicted docked complexes of BID-16 with PSEN-1 and PSEN-2, which showed the lowest binding affinity with least RMSD score. Based on the results of molecular docking simulation, BID's with inferior binding affinity, less number of non-bonding interactions, and hydrogen bonds were selected as negative controls. In case of PSEN-1, BID-19 was selected as a negative control, whereas in case PSEN-2, BID-20 was selected as the same. Molecular dynamics simulation study is done to get the clear insight of the structure and thermodynamics at atomic level, stability, behaviour, and conformational changes of both the protein and the ligand molecule. MD simulation was carried out at nanosecond scales for the docked complex of BID-16 using GROMACS-2018.1 [20] for the duration of 100 ns, to which CHARMM36 force field was applied to get the parameter for protein whereas, CGenFF python script [21] was used to generate the topology and force field parameter file for the ligand. Using TIP3 water model which describes the 3-site rigid water model with charges and Lennard-Jones parameter, a solvent box was created with 10 Å distance. By using genion tool the system was neutralized by maintaining the proper salt concentration of 0.15 M counter ions were added such as $Na^+$ and $Cl^-$. The initial energy minimization step is done for 5000 steps using steepest descent algorithm. Further, the system

equilibration is done using NPT and NVT ensemble class with a 310K temperature and 1 bar pressure. The simulation was run for 100 ns with relaxation time one ps time. After simulations the trajectories were analysed using GROMACS plugin to know the stability of protein and protein-ligand complex system by analysing the geometrical properties, Root Mean Square Deviation (RMSD), Root Mean Square Fluctuation (RMSF), Radius of Gyration (Rg) and SASA (Solvent Accessible Surface Area) were calculated. The plot for the above trajectories was plotted using XMGRACE software [22].

## Binding free energy calculations

The binding free energy of the complex was calculated using Molecular Mechanics/Poisson-Boltzmann Surface Area (MM-PBSA) approach by using g_mmpbsa program [23] which works using GROMACS trajectories. In g_mmpbsa tool the binding free energy is calculated using three components which are molecular mechanical energy, polar and apolar solvation energies. To calculate the binding free energy, the MD trajectories of last 50 ns were considered to compute ΔG with dt 1000 frames. The binding free energy is calculated using below two Eq (I), while the free energy of individual component of the complex is calculate using the Eq (II).

$$\Delta G \text{ binding} = G \text{ complex} - (G \text{ protein} + G \text{ ligand}) \tag{I}$$

$$G = (E_{MM}) - TS + (G_{sol}) \tag{II}$$

$$E_{MM} = E_{bonded} + E_{nonbonded} \tag{III}$$

$$G_{sol} = G_{polar} + G_{nonpolar} \tag{IV}$$

Where $E_{MM}$ is the average potential energy in vacuum, $G_{sol}$ is the sum of the solvation free energy. Further in Eq (III) is given to compute average potential energy in vacuum, the bonded includes bond length, angle and torsion angle where as in nonbonded van der Waals and electrostatic are seen. Eq (IV) is given to find the energy needed to transfer the solute from vacuum to the solvent. The $G_{polar}$ and $G_{nonpolar}$ represents electrostatic and nonelectrostatic support to the solvation free energy.

## ADMET and druglikeliness studies

The ADMET screening was carried out to assess the druglikeliness, pharmacokinetics, and toxicity parameters of ligand molecules through *in silico* approach using SWISS ADME (http://www.swissadme.ch/) and OSIRIS property explorer (http://www.cheminfo.org/Chemistry/Cheminformatics/Property_explorer) [24]. The druglikeliness was evaluated based on "Lipinski's rule of five". Whereas, the toxicity was evaluated based on mutagenic, tumorigenic, irritant, effects on the reproductive system, drug-induced liver injury (DILI) and cytotoxic parameters.

## Results

### Modelled structure validation

The modelled structures were validated for its physiochemical properties using Protparam [25], which depicted the stability, with the instability index of 36.89 for PSEN-1 and 43.45 for PSEN-2. Further, the Ramachandran plot was used to evaluate the quality of the modelled structure based on the Phi and Psi distributions. For PSEN-1 91.5% of the residues were

present in favoured regions and 2.30% of them were in outlier regions (Fig 1). The same pattern was observed with PSEN-2, with 92.7% of the residues present in favoured, and 1.64% of the residues in outlier regions (Fig 2). The overall quality of Ramachandran plots was satisfactory for both the modelled structures. Tables 1 and 2 depict the physiochemical properties of PSEN-1 and PSEN-2 protein models, respectively.

After the validation of modelled structure binding pockets were predicted using CASTp server for PSEN-1 and PSEN-2 proteins (Figs 3 and 4). These results predicted the volume and area of the largest binding pocket, which was found to be 4577.404 Å and 3994.120, respectively, for PSEN-1. Likewise, the volume and area of PSEN-2 are 1560.852 Å and 1554.008 Å. The predicted active site consisting amino acids ranged between 75–435 for PSEN-1, whereas for PSEN-2, it was 82–262.

## Molecular docking simulations

The structures PSEN-1 and PSEN-2, were docked to know the reliable binding interaction and affinity with the BID compounds. The structural differences and results of the docked BID's (binding affinity, total number of non-bounded interactions, and total number of hydrogen bonds) are given in the S1 Table. The best pose for each conformation was chosen based on their binding affinity and RMSD score given by AutoDock Vina 1.1.2. Compound BID-16 showed better interaction with PSEN-1 and PSEN-2 protein (Table 3). The binding affinity of BID-16 with PSEN-1 is of -10.2 kcal/mol, with total number of 12 non-bounded interactions out of which 4 were hydrogen bonds. On the other hand, BID-16 with PSEN-2 results showed the binding energy of -9.4 kcal/mol the total number of 15 non bounded interaction out of which 2 were hydrogen bonds, respectively. The same results of the molecular docking simulation have been represented through column figures for PSEN-1 (Fig 5) and PSEN-2 (Fig 6).

The molecular interaction of BID-16 bound PSEN-1 had total of 12 non bonded interactions, LEU B: 383 (2.94 Å), GLY B: 384 (2.03 Å), ASP B: 385 (3.40 Å), ASP B: 385 (3.75 Å) were bound with hydrogen bonds> In addition 8 hydrophobic bonds were formed, including 2 Pi-sigma bounds with LEU B: 268 (3.93 Å), LEU B: 286 (3.99 Å), 4 alkyl bonds with ALA B: 285 (4.04 Å), ALA B: 434 (4.35 Å), ILE B: 213 (4.39 Å), ILE B: 229 (4.36 Å), and 2 Pi-alkyl bonds with ILE B: 387 (5.06 Å) and ALA B: 285 (4.82 Å). On other hand, BID-16 bond PSEN-2 had total of 15 non bonded interactions, with a total of 2 hydrogen

**Table 1. The predicted values by Protparam server for PSEN-1.**

| | |
|---|---|
| Number of amino acids | 393 |
| Molecular weight | 44096.80 |
| Theoretical pI | 5.70 |
| Instability index | 36.89 |
| Grand average of hydropathicity (GRAVY) | 0.527 |

**Table 2. The predicted values by Protparam server for PSEN-2.**

| | |
|---|---|
| Number of amino acids | 368 |
| Molecular weight | 41261.99 |
| Theoretical pI | 5.01 |
| Instability index | 43.45 |
| Grand average of hydropathicity (GRAVY) | 0.641 |

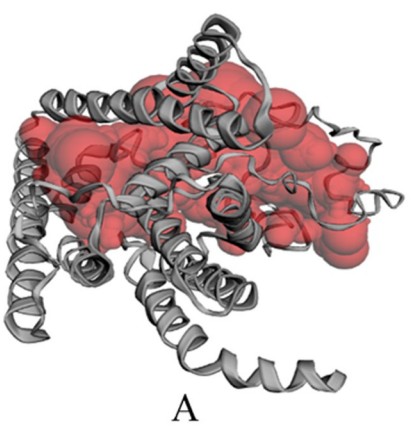

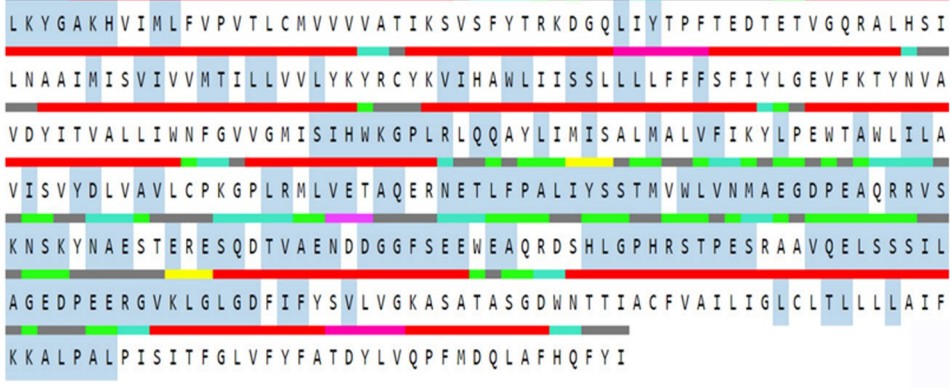

**Fig 3.** CASTp result of PSEN-1 protein A) Binding pocket (highlighted in red) of modelled protein and B) Residues in the sequence (red: α-helix, yellow: strand, pink: π-helix, cyan: turn, green: bend, grey: coil); active binding residues: highlighted in the middle with greyish blue.

bonds with ALA B: 415 (2.26 Å), PRO B: 414 (3.34 Å), 4 electrostatic bonds were formed with ASP B: 263 (5.36 Å), ASP B: 366 (2.89 Å), ASP B: 263 (4.91 Å), ASP B: 366 (3.23 Å). A total of 6 hydrophobic bonds were formed, including a lone Pi- Pi bond with PHE B: 289 (5.16 Å), 2 alkyl bonds with LEU B: 274 (4.28 Å), LEU B: 156 (4.96 Å), and 3 Pi-alkyl bonds with LEU B: 292 (5.10 Å), LEU B: 274 (5.31 Å) and LEU B: 292 (5.46 Å). In case of negative controls, BID-19 formed a total of 9 non-bonding interactions with 1 hydrogen bond with GLY B: 382 (3.74 Å) of PSEN-1. The hydrophobic interactions included pi-sigma bonds with ILE B: 143 (3.41 Å), LEU B: 268 (3.86 Å), and LEU B: 286 (3.63 Å). It formed 3 alkyl bonds with LEU B: 383 (5.19 Å), ILE B: 387 (5.00 Å), and ILE B: 143 (4.53 Å) of PSEN-1. In addition, 2 pi-alkyl bonds with PHE B: 388 (5.40 Å) and LEU B: 286 (4.65 Å) were also formed. With these interactions, the binding affinity of BID-19 was found to be -7.6 kcal/mol. In case of PSEN-2, the BID-20 formed a total of 3 non-bonding interactions with no hydrogen bonds. It formed a hydrophobic bond with LEU B: 274 (3.53 Å), and 2 pi-alkyl bonds with ILE B: 288 (4.89 Å) and ILE B: 293 (5.33 Å). With these interactions, the binding interaction of BID-20 is -8.2 kcal/mol. The visualization of molecular docking for PSEN-1 and PSEN-2 with BID's has been given in **Figs 7** and **8**, respectively.

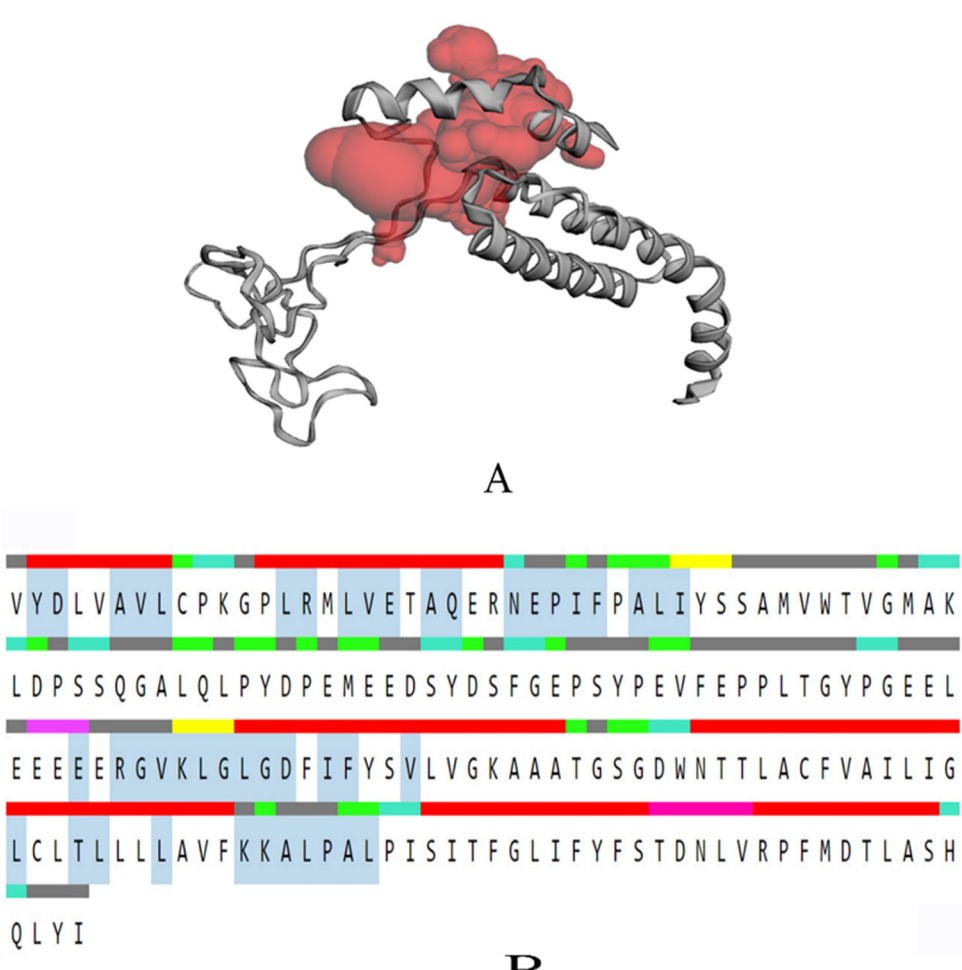

**Fig 4.** CASTp result of PSEN-2 protein A) Binding pocket (highlighted in red) of modelled protein and B) Residues in the sequence (red:α-helix, yellow: strand, pink: π-helix, cyan: turn, green: bend, grey: coil); Active binding residues: highlighted in the middle with greyish blue.

### Druglikeliness and pharmacokinetics analyses

The Lipinski's rule of five is used to determine the pharmacokinetic potential of the chemical compounds through the means of their absorption, distribution, metabolism, elimination (ADMET) properties. During this analysis, the drug score predicted by OSIRIS for BID-16 was 0.76. Further, the pharmacokinetic properties were predicted with the molecular weight 432.32 g/mol, number of hydrogen bond acceptors 7, number of hydrogen bond donors 2, number of rotatable bonds 7 and cLogP 3.18. There were no toxicity issues like mutagenicity, tumerigenicity, irritability reported with BID-16. Furthermore, it was predicted with no inhibition of P-glycoprotein (P-gp) substrate, cytochrome P (CYP). In case of negative controls, the druglikeliness of depict that the molecular weights of BID-19 and BID-20 is found to be 477.63 g/mol and 522.08 g/mol, respectively. Both the compounds had same hydrogen bond acceptors (3), donors (0), and rotatable bonds (6). However, the cLogP value BID-19 (6.07) and BID-20 (6.18) differed. Regarding pharmacokinetic properties, tumerigenicity was found in case of BID-19 as well as BID-20. Both of the compounds didn't pass druglikeness parameter (-2.95 for BID-19) and (-2.45 for BID-20), as both of them violate the Lipinski's rule of five.

**Table 3. Binding interaction formed with their respective residues formed during docking of BID-16, BID-19, and BID-20 with PSEN-1 and PSEN-2 (distances shown in brackets are measured in Å).**

| Sl. No. | Name of the compound | Binding affinity (kcal/mol) | Hydrogen bonds | Electrostatic bonds | Hydrophobic bonds | | | |
|---|---|---|---|---|---|---|---|---|
| | | | | | Pi-sigma | Pi- Pi bond | Alkyl | Pi-alkyl |
| 1 | PSEN-1 with BID-16 | -10.2 | LEU B: 383 (2.94), GLY B: 384 (2.03), ASP B: 385 (3.40), ASP B: 385 (3.75) | - | LEU B: 268 (3.93), LEU B: 286 (3.99) | - | ALA B: 285 (4.04), ALA B: 434 (4.35), ILE B: 213 (4.39), ILE B: 229 (4.36) | ILE B: 387 (5.06), ALA B: 285 (4.82) |
| 2 | PSEN-1 with BID-19 | -7.6 | GLY B: 382 (3.74) | | ILE B: 143 (3.41), LEU B: 268 (3.86), LEU B: 286 (3.63) | | LEU B: 383 (5.19), ILE B: 387 (5.00), ILE B: 143 (4.53) | PHE B: 388 (5.40), LEU B: 286 (4.65) |
| 3 | PSEN-2 with BID-16 | -9.4 | ALA B: 415 (2.26), PRO B: 414 (3.34) | ASP B: 263 (5.36), ASP B: 366 (2.89), ASP B: 263 (4.91), ASP B: 366 (3.23) | - | PHE B: 289 (5.16) | LEU B: 274 (4.28), LEU B: 156 (4.96) | LEU B: 292 (5.10), LEU B: 274 (5.31), LEU B: 292 (5.46) |
| 4 | PSEN-2 with BID-20 | -8.2 | - | - | LEU B: 274 (3.53), | | - | ILE B: 288 (4.89), ILE B: 293 (5.33) |

## Molecular dynamics simulations

The molecular dynamics simulation was performed to evaluate the conformational changes and dynamic stability of the docked complexes. The MD was performed for homology models of PSEN-1 and PSEN-2 with the selected compound of BID-16 as their ligand. The simulation was carried for 100 ns to understand the interactions between the docked complexes and the results were evaluated based on, RMSD, Rg, RMSF, SASA and ligand hydrogen bond plots.

The RMSD was evaluated to know the structural stability of both the protein and ligand complex and to examine their dynamic nature during the 100 ns simulation. The values were evaluated based on the plot of PSEN-1 and PSEN-2 with BID-16. The RMSD of PSEN-1-BID-16 complex that covered during 100 ns MD simulation showed that the complex attained stability after 80 ns (**Fig 9A**). The PSEN-1 protein (green) increased up to 0.6 nm at ~5 ns and there was a slight fluctuation around 50 to 55 ns at 0.62 nm. Later, it demonstrated stability over ~78 to 80 ns and a slight dip of 0.58 ns at 80 ns, whereas protein-BID-16 complex (black) increased at a time interval of 0.5 ns at 0.55 nm. Further, there a slight rise at ~10 ns of 0.59 ns was observed that remained stable throughout the run. However, protein-BID-19 complex became stable after 90 ns. It rose to 0.8 nm at 5 ns, and the peak value (0.9 nm) was observed at

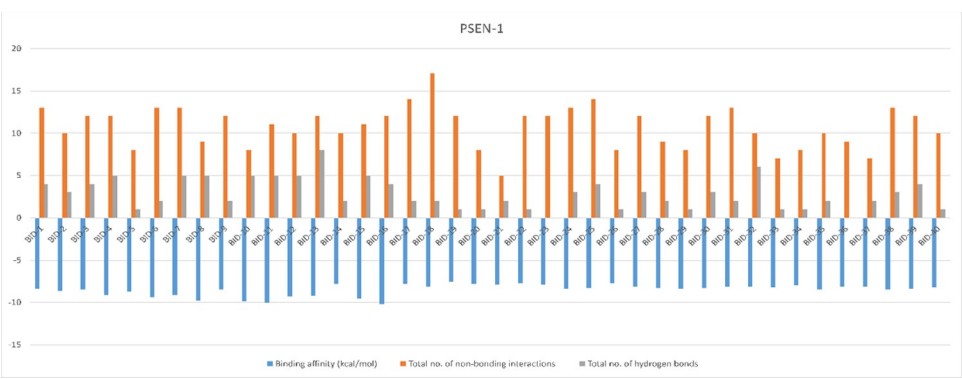

**Fig 5. Binding affinity, non-bonding interactions, and hydrogen bonds formed during the molecular docking simulation of PSEN-1.**

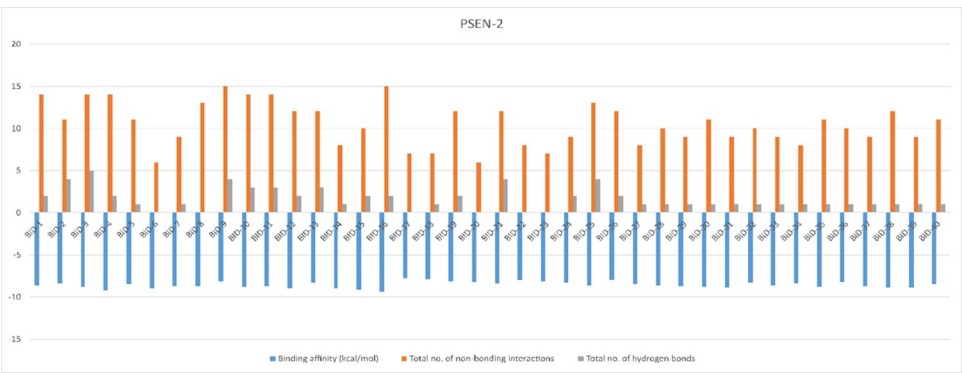

**Fig 6. Binding affinity, non-bonding interactions, and hydrogen bonds formed during the molecular docking simulation of PSEN-2.**

20 ns, 50 ns, and 70 ns. The whole RMSD plot of protein-BID-19 complex was found to be unstable throughout the simulation, in comparison to the protein-BID-16 complex. On the other hand, PSEN-2-BID-16 complex attained stability around 90 ns at 0.98 ns (**Fig 10A**). The PSEN-2 (green) increased up to 1 nm at around 45 ns time and remained its stability throughout the simulation, whereas the protein-BID-16 complex (black) showed a slow increase up to 0.98 nm at ~35 ns, then fell to 0.9 nm at the time interval of 50 ns and remained stable through 100 ns run. The protein-BID-20 complex however, showed constant fluctuations throughout the simulation run. The peak of RMSD plot was observed at 40–45 ns (1.5 nm). Unlike protein-BID-16 complex, it was not stable from the beginning to the end.

The RMSF plot was analysed to evaluate the residue flexibility of complex along with the modelled protein. The RMSF was evaluated for both PSEN-1 and PSEN-2 along with the BID-16 complex as depicted in the **Figs 9B** and **10B**, respectively. The RMSF plot of PSEN-1 model

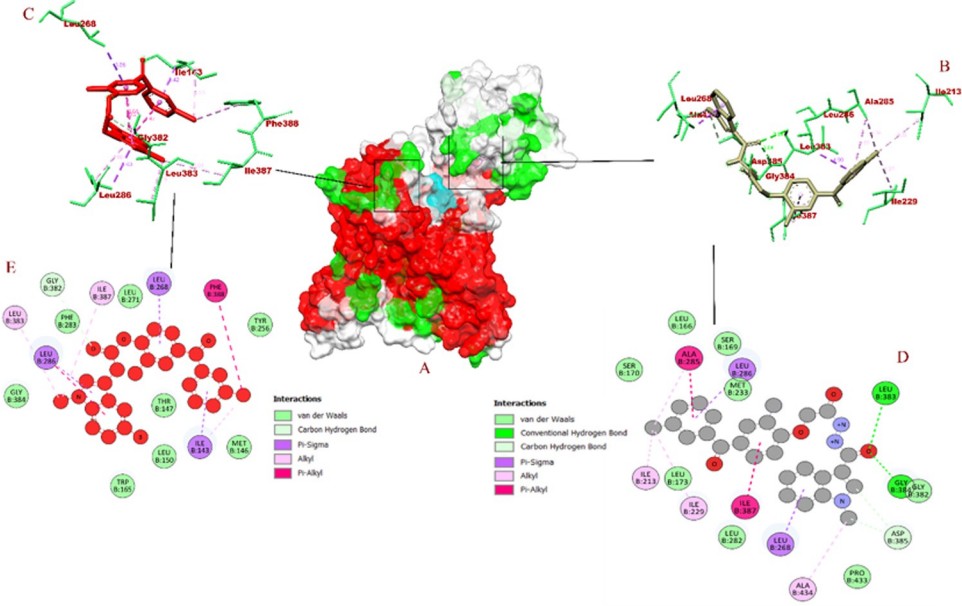

**Fig 7.** Visualization of docking interaction of BID-16 and BID-19 (negative control) bound with PSEN-1 protein (A) atom type surface (B) 3D binding interaction of BID-16 (C) 3D binding interaction of BID-19 (highlighted in red) (D) 2D binding interaction of BID-16 (E) 2D binding interaction of BID-19 (highlighted in red).

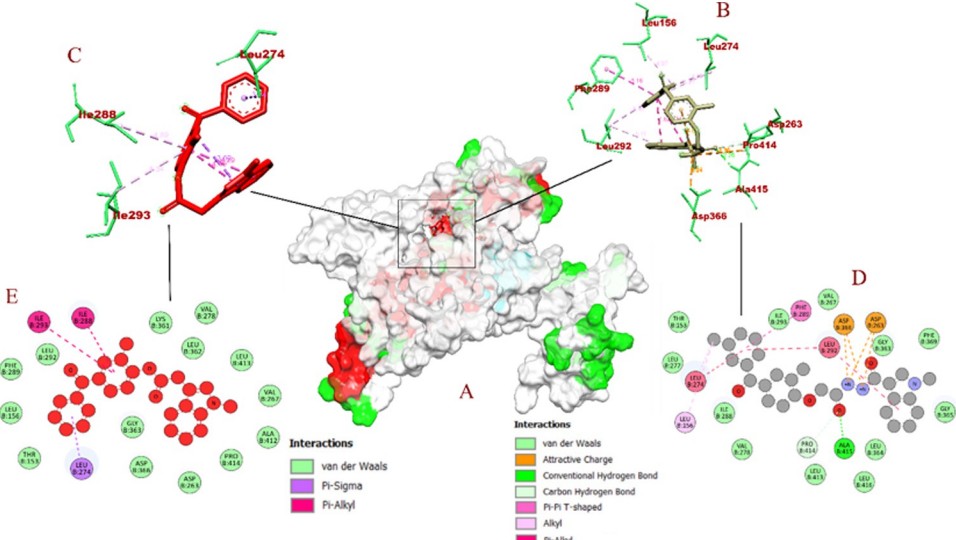

**Fig 8.** Visualization of docking interaction of BID-16 and BID-20 (negative control) bound with PSEN-2 protein (A) atom type surface (B) 3D binding interaction of BID-16 (C) 3D binding interaction of BID-20 (highlighted in red) (D) 2D binding interaction of BID-16 (E) 2D binding interaction of BID-20 (highlighted in red).

showed a highest fluctuation at T-terminal for the residues at the 450 position at 1.5 nm, whereas the protein-BID-16 was also predicted with the same value. The protein-BID-19 complex was found with unusual fluctuations at 250–400 residues, indicating its relative instability. It was also found with highest RMSF value at T-terminal (1.5 nm). Conversely, PSEN-2 showed a highest fluctuation at loop regions 100–150 and 400 residues. Otherwise, it was found with minimal fluctuations. The protein-BID-16 complex was also found with minimal fluctuations, except loop regions. However, the protein-BID-20 complex was found with higher number fluctuations at loop regions compared to protein backbone atoms and protein-BID-16 complex, indicating its instability. It showed the highest fluctuation at 300–350 residues.

The Rg was analysed to evaluate the structural compactness of the bio-molecules and to check if the complexes are stably folded or unfolded. The average Rg value of PSEN-1-BID-16 complex (**Fig 9C**) was 2.24 nm, followed by a slight reduction in the fluctuation value at 2.24 nm at 40 ns and showed relatively stable values thereafter (**Fig 10C**). The protein as well as the complexes showed almost similar and compactable values for PSEN-1-BID-16 complex. Based on the above evaluation, it can be suggested that the PSEN-1-BID-16 complex may show relatively stable folded conformation during MD simulation. In case of PSEN-2, the same pattern was followed, where the Rg plot of protein-BID-16 was found to be stable in comparison with that of the protein-BID-20 complex. The **Figs 9D** and **10D** shows the graphical plot of solvent accessible surface area (SASA) through which conformational changes between the interaction is analysed. The plot was evaluated based on the SASA values with simulation time i.e. 100 ns for both the modelled protein with BID-16. The average value of PSEN-1 with BID-16 was 181 nm$^2$. Whereas, the protein-BID-19 complex was found to in the same pattern. In case of PSEN-2, both PSEN-2 bound BID-16 and BID-20 showed 160 nm$^2$ of SASA value. Finally, the number of hydrogen bonds which was formed during the interaction were calculated. In this analysis, BID-16 showed up to 8 hydrogens bonds, whereas BID-19 showed only 3 of them (**Fig 9E**). In case of PSEN-2, BID-16 was found with 7 hydrogen bonds, where BID-20 was able to show 5 hydrogen bonds (**Fig 10E**). The ligand hydrogen bonds depict the relatively

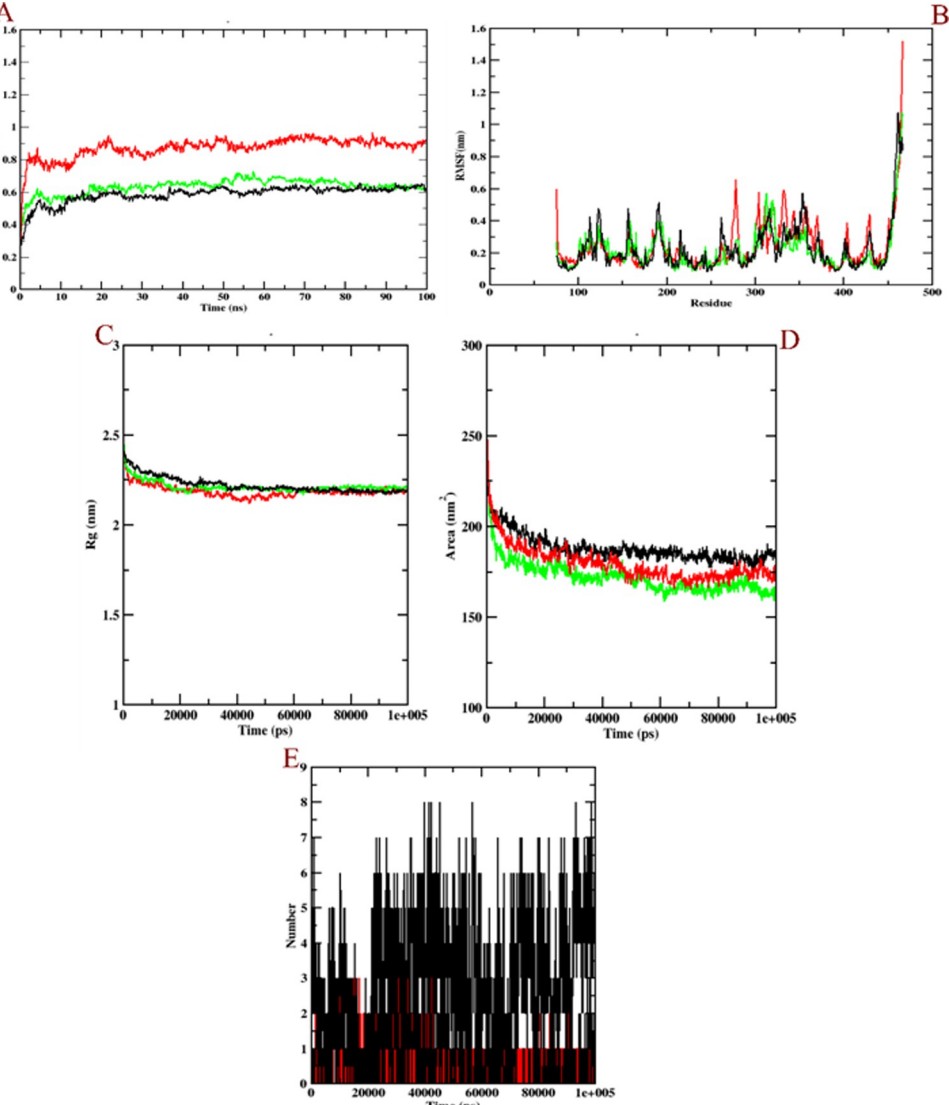

**Fig 9. Plot of the molecular dynamics simulations trajectories obtained after 100 ns interaction for BID-16 and BID-19 (negative control) bound with PSEN-1 protein.** (A) RMSD (B) RMSF (C) Rg (D) SASA, and (E) ligand hydrogen bonds; Green: protein backbone atoms, black: protein-BID-16 complex, red: protein-BID-19 complex (negative control).

better binding interaction of BID-16 compared to BID-19 and BID-20 during dynamics simulation. The hydrogen bond varies from docked model to MD simulated model which shows that there is a structural rearrangement.

## Binding free energy calculations

Evaluation of binding free energy was done using MM-PBSA, which is the summation of non-polar, polar and non-bonded interaction energies. The summary of the free binding free energy calculations of BID-16 and controls (BID-19 and BID-20) bound to PSEN-1 and PSEN-2 have been given in **Table 4**. Compound BID-16 was predicted to form the protein-ligand complex majorly using the Van der Waal's energy in case of both PSEN-1 and 2. This was followed by binding energy, which followed the same pattern as the Van der Waal's for

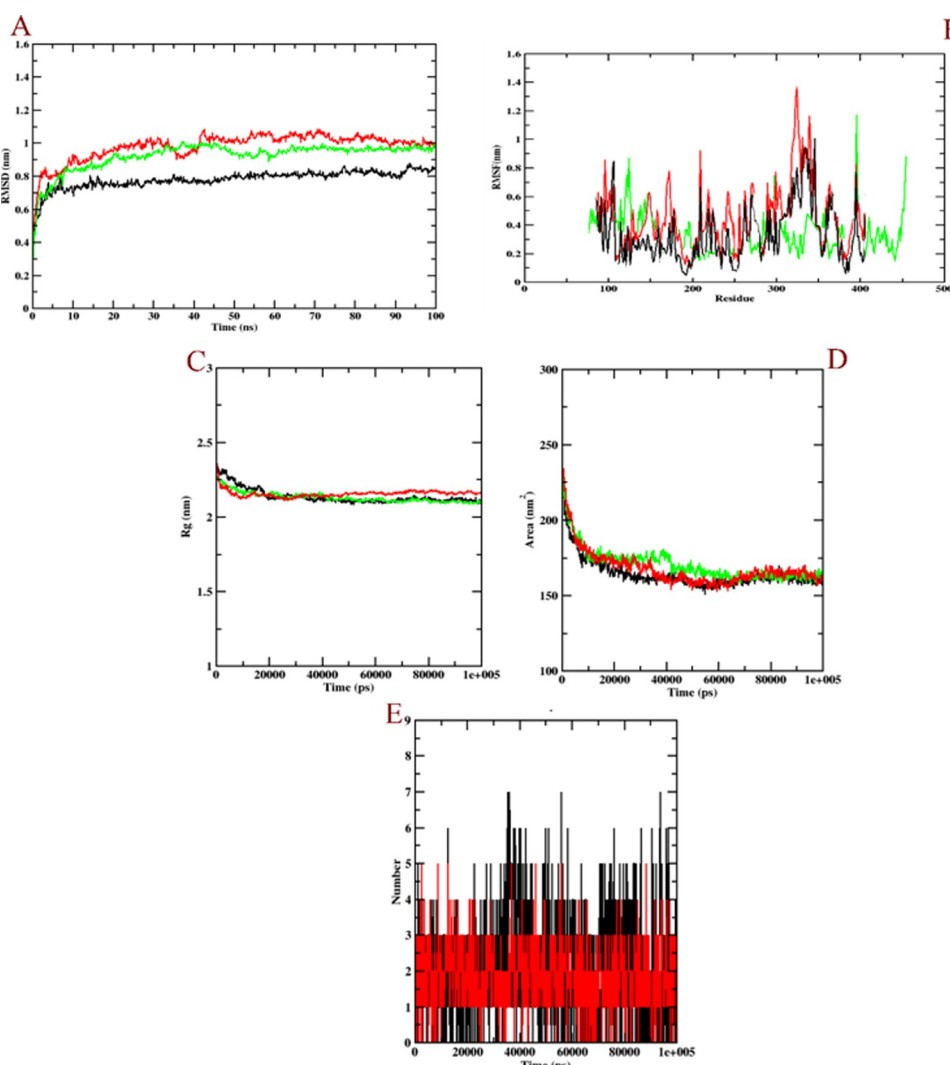

**Fig 10. Plot of the molecular dynamics simulations trajectories obtained after 100 ns interaction for BID-16 and BID-20 (negative control) bound with PSEN-2 protein.** (A) RMSD (B) RMSF (C) Rg (D) SASA, and (E) ligand hydrogen bonds; Green: protein backbone atoms, black: protein-BID-16 complex, red: protein-BID-20 complex (negative control).

**Table 4. Binding free energy analysis of BID-16 and controls bound with PSEN-1 and PSEN-2.**

| Types of binding free energy | PSEN-1-BID-16 complex | | PSEN-1-BID-19 complex | | PSEN-2-BID-16 complex | | PSEN-2-BID-20 complex | |
|---|---|---|---|---|---|---|---|---|
| | Values (kj/ mol) | Standard deviation (kj/mol) | Values (kj/ mol) | Standard deviation (kj/mol) | Values (kj/ mol) | Standard deviation (kj/mol) | Values (kj/ mol) | Standard deviation (kj/mol) |
| Van der Waal energy | -251.413 | +/- 151.030 | -264.128 | +/-120.671 | -341.957 | +/- 26.959 | -410.531 | +/-209.673 |
| Electrostatic energy | -15.695 | +/- 12.775 | -17.578 | +/-10.352 | -36.231 | +/- 12.299 | -41.391 | +/-11.459 |
| Polar solvation energy | 126.650 | +/- 77.941 | 132.435 | +/-81.231 | 159.179 | +/- 17.210 | 164.465 | +/-100.539 |
| SASA energy | -18.186 | +/- 11.003 | -21.451 | +/-11.562 | -22.367 | +/- 1.011 | -31.917 | +/-2.139 |
| Binding energy | -158.644 | +/- 101.666 | -169.105 | +/-112.081 | -241.376 | +/- 24.462 | -313.128 | +/-198.871 |

both the proteins. However, polar solvation energy did not contribute for the formation of protein-BID-16 complex, as it was predicted with positive values in case of both the proteins. Compared to complexes formed with BID-16, negative controls (complexes formed with BID-19 and BID-20) were found with more standard deviations. The higher standard deviations depict the inconsistent and unstable binding interactions during molecular dynamics simulation.

## Discussion

Benzophenone derivatives are known to be promising treatment for AD because of their acetylcholinesterase (AChE) inhibiting potential, which helps in increasing the level of the neurotransmitter actions [5, 25]. Accumulation of Aβ peptide primarily due to the increased level of AChE effects the plaques deposition leading to the pathological condition [9]. So far, there are several classes of AChE inhibitors, some of which are said to be FAD approved drugs commercially available. These include donepezil, rivastigmine, and galantamine, which can only slow the progression of AD but cannot prevent or completely treat the condition.

In this study, we use *in-silico* approach to investigate the effect of benzophenone based compounds out of which BID-16 was chosen after the virtual screening against PSEN-1 and PSEN-2 targets to know the effect on AD and to evaluate the biological activity, pharmacological effect, binding interaction, and dynamic nature on the target proteins. To analyse the molecular interaction of the compounds with modelled PSEN-1 and PSEN-2 proteins, docking study was conducted, after which the compound BID-16 was chosen for further investigation as it showed better binding affinity of -10.2 kcal/mol for PSEN-1 and -9.4 kcal/mol for PSEN-2, which were considered as the lowest negative values. The lowest negative values indicate the better binding affinity and interaction. To better understand the interaction patterns, the Discovery Studio Visualizer 2021 was used to analyze the docked conformations. The evaluation of PSEN-1 and PSEN-2 bound to BID-16 showed that it is bound within the binding site of both the proteins.

The binding interaction of BID's with both PSEN-1 and PSEN-2 have been analysed *in silico*. The UniProt sequence and binding pocket analysis of PSEN-1 depict that 2 of the catalytic residues (ASP B: 257 and ASP B: 385) are involved in the inactivation of the enzyme activity, if the inhibitor binds to them [26, 27]. Since PSEN-2 being a functionally similar and homologous to PSEN-1 [28], binding of inhibitor molecules to the binding pocket could induce the protein inactivity.

A study by **Yang et al. (2021)** states that both PSEN-1 and PSEN-2 are the fuctional subunits of γ-secretase (PDB ID: 7C91). The study also states that ASP B: 257 and ASP B: 385 are the catalytic residues of the PSEN-1 present in the γ-secretase. When they investigated the inhibition activity of avagacestat, the compound bound to LEU B: 268 [29]. In our study, BID-16 was able to bind with both ASP B: 385 and LEU B: 268 (**Table 3**). With these studies, we can conclude that BID-16 could be a potent inhibitor of PSEN-1.

However, another γ-secretase inhibitor drug known as LY450139, is in Phase III clinical trial [30]. When we performed molecular docking simulation, LY450139 was able to bind with our homology built models of PSEN-1 (**S1 Fig**) and PSEN-2 (**S2 Fig**) *in silico*. The docking results depict that ALA 415 of PSEN-2 is a common binding residue of LY450139 and BID-16. The LY450139 and BID-16 were able to bind with GLY B: 384 (**S2 Table**) and ASP 385 (**Table 3**) of PSEN-1, respectively. The binding interactions of LY450139 have been depicted in **S2 Table**. Moreover, our study highlights the binding of BID-16 to the same binding pocket of the PSEN-1 and PSEN-2 that has been highlighted in the above mentioned studies. With

these outcomes, we can conclude that BID's can act the potential inhibitors of both the presenilin proteins.

The BID-16 was taken for further investigation to assess its druglikeliness and pharmacokinetics properties, and the total binding energy formed during the dynamic simulation. The drug score of BID-16 was found to be 0.76. The compound passed the Lipinski's rule of five, as it had all the essential properties. Likewise, there was no liver cytotoxicity, which is one of the most important parameter for drugs suitability evaluation. The compound demonstrated no Cytochrome P and p-gp inhibition, suggesting its potential as a promising drug candidate. Therefore, BID-16 can be considered to have better oral absorption.

Furthermore, the molecular dynamic study was conducted to know the stability and structural changes of the protein-ligand complexes. In this study, we performed a MDS run for 100 ns for the complexes along with its reference compounds. The RMSD, RMSF, Rg, SASA and ligand hydrogen bonds were analysed using MD trajectories. All the results from MD analysis depict BID-16 as a promising drug candidate in terms of its stability and extensive binding interaction inside the binding pocket of the proteins for 100 ns. However, the changes observed in the trajectory plots reveal that interaction of BID-16 with both the proteins has been exceptional in terms of overall stability, in comparison with the negative controls. The values of RMSD, RMSF, Rg, SASA and ligand hydrogen bonds for protein-BID-16 complex are in accordance with those of protein backbone atoms. However, proteins complexed with negative controls showed inferior results. In addition, the binding free energy calculations showed that BID-16 bound PSEN-1 and PSEN-2 had lower binding free energies, indicating that the binding efficiency improved over the time frame. Also, standard deviations depict that the molecular dynamics simulation done for the protein-BID-16 has been remarkable in terms its stability, in comparison with the negative controls.

The hallmark of this study is to investigate the presenilin inhibitory potential of BID's through *in silico* approach. Till date, there have been no studies in this aspect. So far, this is the first computational investigation on PSEN-1 and PSEN-2 proteins, which provides an insight into the mechanism of the inhibition by BID's.

## Conclusion

The main framework of this study was to understand the interaction of benzophenone integrated derivatives with presenilin (PSEN) proteins using computational tools. The molecular docking and interaction study showed that the compound BID-16 demonstrated better binding affinity for both presenilin (PSEN) proteins. Further, pharmacological potential and toxicity analysis was conducted, which screened the BIDs and predicted BID-16 as the most potent inhibitor. Finally, molecular dynamics showed that BID-16 is stable and has better affinity inside the binding pocket of both the proteins. To get a better understanding of the free energy formed during the simulation, binding free energy were calculated, which depicted the role of Van der Waal's and binding energies in the formation of PSEN and BID-16 complexes. Based on these results, it can be deduced that BID-16 is a potential drug candidate for presenilin protein activity inhibition. However, further validation of BID-16 is needed with respect to *in vitro* and *in vivo* aspects to prove its potential as an anti-Alzheimer's drug.

## Supporting information

**S1 Fig. Visualization of binding interactions of LY450139 with PSEN-1.**
(DOCX)

**S2 Fig. Visualization of binding interactions of LY450139 with PSEN-2.**
(DOCX)

**S1 Table. Structural differences and results of molecular docking simulation of BID's with PSEN-2 and PSEN-2 proteins.**
(DOCX)

**S2 Table. Binding affinity and interactions of LY450139 with PSEN-1 and PSEN-2.**
(DOCX)

## Acknowledgments

Authors are thankful to the JSS AHER for their kind support and necessary facilities.

## Author Contributions

**Conceptualization:** Ramith Ramu.

**Data curation:** Shaukath Ara Khanum, Ekaterina Silina.

**Formal analysis:** Reshma Mary Martiz, Shashank M. Patil, Jayanthi M. K.

**Investigation:** Ramith Ramu, Ashwini P.

**Methodology:** Reshma Mary Martiz, Raghu Ram Achar.

**Resources:** Lakshmi V. Ranganatha, Ekaterina Silina, Raghu Ram Achar.

**Software:** Shashank M. Patil, Ashwini P., Lakshmi V. Ranganatha, Shaukath Ara Khanum, Victor Stupin.

**Supervision:** Shaukath Ara Khanum.

**Validation:** Ashwini P., Lakshmi V. Ranganatha, Shaukath Ara Khanum, Ekaterina Silina, Victor Stupin, Raghu Ram Achar.

**Visualization:** Ekaterina Silina, Victor Stupin, Raghu Ram Achar.

**Writing – original draft:** Shashank M. Patil.

**Writing – review & editing:** Ramith Ramu.

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
