## [Decision Letter · Decision Letter 0]

13 Dec 2021

PONE-D-21-34824Discovery of novel benzophenone integrated derivatives as anti-Alzheimer’s agents targeting presenilin-1 and presenilin-2 inhibition: a computational approachPLOS ONE

Dear Dr. Ramu,

Thank you for submitting your manuscript to PLOS ONE. After careful consideration, we feel that it has merit but does not fully meet PLOS ONE’s publication criteria as it currently stands. Therefore, we invite you to submit a revised version of the manuscript that addresses the points raised during the review process.

We look forward to receiving your revised manuscript.

Kind regards,

Jie Zheng, Ph.D

Academic Editor

PLOS ONE

Journal Requirements:

The author(s) reported there is no funding associated with the work featured in this article.

The author(s) reported there is no funding associated with the work featured in this article

4. Please amend your authorship list in your manuscript file to include author Reshma Mary Martiz and Shashank M Patil.

5. We note that Figures 1 and 5 in your submission contain copyrighted images. All PLOS content is published under the Creative Commons Attribution License (CC BY 4.0), which means that the manuscript, images, and Supporting Information files will be freely available online, and any third party is permitted to access, download, copy, distribute, and use these materials in any way, even commercially, with proper attribution. For more information, see our copyright guidelines: http://journals.plos.org/plosone/s/licenses-and-copyright.

a. You may seek permission from the original copyright holder of Figures 1 and 5 to publish the content specifically under the CC BY 4.0 license. 

Reviewers' comments:

Reviewer's Responses to Questions

**Comments to the Author**

1. Is the manuscript technically sound, and do the data support the conclusions?

Reviewer #1: Partly

Reviewer #2: Partly

2. Has the statistical analysis been performed appropriately and rigorously? 

Reviewer #1: No

Reviewer #2: Yes

3. Have the authors made all data underlying the findings in their manuscript fully available?

Reviewer #1: Yes

Reviewer #2: Yes

4. Is the manuscript presented in an intelligible fashion and written in standard English?

Reviewer #1: Yes

Reviewer #2: Yes

5. Review Comments to the Author

Reviewer #1: The study analyzed the binding affinity between benzophenones and PSEN-1/2 for demonstrating the pharmacotherapeutic potential by inhibiting PSEN-1/2. However, the study only focused on analyzing the affinities and interactions, without further investigation of the preferential binding pocket for the inhibition effect. Herein, it is hard to conclude the inhibition efficiency of screened benzophenone from these preliminary data, and many issues need to be addressed as followings:

1. Figure 3-4, the binding pocket and binding sequence labeled in different colors should be clearly explained in the figure caption.

2. Table 3-4, the in parallel comparison of binding affinity and non-bonding interactions of various BIDs are encouraged to present in column figures for better illustration. Also, please highlight the molecule structural differences of these BIDs in different color.

3. Figure 5-6, it is hard to distinguish between the Pi-alkyl and alkyl interactions.

4. The binding interactions between BIDs and PSEN-1/2 were analyzed in silico, but for the inhibition effect, how this binding contribute to the inhibition? Whether by stabilize the PSEN-1/2 conformation or block the active sites? All these need to be further introduced to demonstrate the inhibition benefits.

5. In addition to the binding affinity, the binding pocket should also be analyzed for screening the best pose of BID-16 with PSEN-1/2.

Reviewer #2: In the manuscript PONE-D-21-34824, the study evaluates the interaction of BIDs through molecular docking simulations, molecular dynamics simulations, and binding free energy calculations. Combinations of these approaches are acceptable. Although the manuscript is informative, there are still some questions that need to be addressed (see comments listed below).

1. The false-positive rate of molecular docking is very high in the absence of experimental support. The literature listed by the author only mentions the inhibition of AChE/BACE-1 by Benzophenones, not PSEN-1/PSEN-2. Therefore, there is little significance in performing molecular docking study in such cases.

2. The authors used CHARMM27 force field to get the parameter for protein (Line 126 to 129 in manuscript). However, their studies were based on an obsolete force field, which need a revision with the latest force fields. This can impact the results, weakening some analysis. One should use CHARMM36 instead of 27 (please see doi.org/10.1021/ct300400x).

3. Typically, the two compounds with the best activity and the worst activity are usually selected for comparison in MD study. The authors should add a compound as the negative control.

4. Table 3 and Table 4 contain the same dataset and should be merged into one table.

5. One gets the impression that the work was assembled very hastily and without careful control. For instance, there are miscellaneous lines on the Fig.8B, which may be caused by the incorrect modification of the residue number. Also, Fig. 7B and 8B are missing ordinate labels. The units in the pictures should be unified (e.g. Fig7A, 7C, 7D, 7E, 8A, 8C, 8D, 8E). As a suggestion, the authors should use professional drawing software to improve the quality of these pictures.

6. PLOS authors have the option to publish the peer review history of their article (what does this mean?). If published, this will include your full peer review and any attached files.

Reviewer #1: No

Reviewer #2: No

---

## [Author Response · Author response to Decision Letter 0]

18 Jan 2022

Reviewer 1 comments

The study analysed the binding affinity between benzophenones and PSEN-1/2 for demonstrating the pharmacotherapeutic potential by inhibiting PSEN-1/2. However, the study only focused on analysing the affinities and interactions, without further investigation of the preferential binding pocket for the inhibition effect. Herein, it is hard to conclude the inhibition efficiency of screened benzophenone from these preliminary data, and many issues need to be addressed as followings:

1. Figure 3-4, the binding pocket and binding sequence labelled in different colours should be clearly explained in the figure caption.

Response: As per the reviewer’s suggestion, binding pocket and binding sequence labelled in different colours (for both PSEN-1 and PSEN-2) have been explained in figure captions of Figure 3 (line 201-204) and Figure 4 (line 206-209).

2. Table 3-4, the in parallel comparison of binding affinity and non-bonding interactions of various BIDs are encouraged to present in column figures for better illustration. Also, please highlight the molecule structural differences of these BIDs in different colour.

Response: As per the reviewer’s suggestion, Table 3 and Table 4 have been merged. The merged table (Table 3, line 225) now contains the structural differences of the BIDs highlighted in different colour. Also, binding affinity, total number of non-bonding interactions, total number of hydrogen bonds formed by BIDs during the docking simulation have also been given in adjacent columns. For the better illustration of these interaction parameters, Figure 5 and Figure 6 have been added, which represent the different docking parameters of PSEN-1 and PSEN-2, respectively.

3. Figure 5-6, it is hard to distinguish between the Pi-alkyl and alkyl interactions.

Response: As per the reviewer’s suggestion, colour of the Pi-alkyl interactions have been changed to dark pink colour (now Figure 7 and Figure 8). Also, row headings have been added in Table 4, which clearly describes the types of interactions to avoid confusion.

4. The binding interactions between BIDs and PSEN-1/2 were analysed in silico, but for the inhibition effect, how this binding contributes to the inhibition? Whether by stabilize the PSEN-1/2 conformation or block the active sites? All these need to be further introduced to demonstrate the inhibition benefits. 

Response: As per the reviewer’s suggestion, in silico interaction of BIDs have now been supported with suitable information in “Discussion” section (lines 439-462) and supplementary data. In order to do this, authors have carefully performed the binding pocket analysis of both PSEN proteins, followed by the interaction analysis of BIDs with specific amino acid residues. We have identified a few key amino acid residues from the literature analysis, binding to which induces the inhibition activity. Binding interactions of BIDs also reveal that the key amino acids have been bound with the BIDs. Therefore, the interaction is oriented towards the inhibition of the protein activity by blocking the active binding residues.

From previous studies conducted on γ-secretase, which consists of PSEN-1 and PSEN-2 proteins as functional subunits, we have provided the information of in silico inhibition of γ-secretase. The same inhibitor compound (LY450139) used in the study was docked with our homology model built PSEN-1 and PSEN-2 proteins. This has produced similar binding interactions like BIDs, which depicts that the binding interactions of BIDs have been oriented towards the inhibition, not the activation of the target proteins.

5. In addition to the binding affinity, the binding pocket should also be analysed for screening the best pose of BID-16 with PSEN-1/2.

Response: As per the reviewer’s suggestion, a thorough analysis of binding pockets of both PSEN-1 and PSEN-2 was conducted using the literature given in the UniProt database website. The literature survey resulted in the identification of few key amino residues, binding to which causes the inhibition of the protein activity. Results from our molecular docking studies reveal that BID-16 binds to the same amino acid residues, which shows that BID-16 as a potential inhibitor of both the PSEN-1 and PSEN-2 proteins. Difference in MD simulation plots were also given to highlight the impact of interaction (lines 476-481). During the virtual screening of the different binding poses, the first binding pose with zero RMSD and most negative binding affinity was selected. The same binding pose was found to interact with the key residues, in comparison with the other 9 binding poses.

Reviewer 2 comments

In the manuscript PONE-D-21-34824, the study evaluates the interaction of BIDs through molecular docking simulations, molecular dynamics simulations, and binding free energy calculations. Combinations of these approaches are acceptable. Although the manuscript is informative, there are still some questions that need to be addressed (see comments listed below).

1. The false-positive rate of molecular docking is very high in the absence of experimental support. The literature listed by the author only mentions the inhibition of AChE/BACE-1 by Benzophenones, not PSEN-1/PSEN-2. Therefore, there is little significance in performing molecular docking study in such cases. 

Response: As per the reviewer’s suggestion, we conducted an extensive literature survey on the involvement of BIDs in the induction of anti-Alzheimer’s activity. However, there have been no studies performed in vitro for the inhibition of PSEN-1 and PSEN-2 by BIDs. The only available study on the anti-Alzheimer’s activity of BIDs is the inhibition of AChE/BACE-1 by Belluti et al. (2014) (reference no. 5). Therefore, we aimed to investigate the anti-Alzheimer’s activity through PSEN inhibition. This study provides an insight to the molecular mechanism of inactivating PSEN proteins through blocking their active sites. As this study is an in silico investigation, in vitro and in vivo studies in future could be performed using BIDs aiming for PSEN inhibition.

2. The authors used CHARMM27 force field to get the parameter for protein (Line 126 to 129 in manuscript). However, their studies were based on an obsolete force field, which need a revision with the latest force fields. This can impact the results, weakening some analysis. One should use CHARMM36 instead of 27 (please see doi.org/10.1021/ct300400x).

Response: As per the reviewer’s suggestion, the force field used for molecular dynamics simulation was changed from CHARMM27 to CHARMM36. According to the reviewer 2 comments the negative control was need to be added in case of both docking and dynamics simulation processes. Therefore, MD simulations of all the BIDs for both PSEN-1 and PSEN-2 were performed again using CHARMM36 force field.

3. Typically, the two compounds with the best activity and the worst activity are usually selected for comparison in MD study. The authors should add a compound as the negative control.

Response: As per the reviewer’s suggestion, negative controls with worst activity were selected for the comparative study. In case of PSEN-1, BID-19 was selected as the negative control, whereas in case of PSEN-2, BID-20 was selected as the same. The visualization/representation of docking studies, dynamics simulation studies, binding free energy calculations, druglikeliness, and pharmacokinetic analysis were also completed by comparing the BID-16 with negative controls.

4. Table 3 and Table 4 contain the same dataset and should be merged into one table.

Response: As per the reviewer’s suggestion, both Table 3 and Table 4 were merged into one table (now Table 3). The table now contains data of structural differentiation of BIDs as well as their interaction details.

5. One gets the impression that the work was assembled very hastily and without careful control. For instance, there are miscellaneous lines on the Fig.8B, which may be caused by the incorrect modification of the residue number. Also, Fig. 7B and 8B are missing ordinate labels. The units in the pictures should be unified (e.g. Fig7A, 7C, 7D, 7E, 8A, 8C, 8D, 8E). As a suggestion, the authors should use professional drawing software to improve the quality of these pictures.

Response: As per the reviewer’s suggestion, negative controls were added for the MD simulation study. Using CHARMM36 forcefield, all the simulations were re-run for 100 ns. All the trajectories of MD simulations were plotted using XMGRACE. The plots were unified and put in one single picture using InkScape software.

---

## [Editor Report · Decision Letter 1]

21 Feb 2022

Discovery of novel benzophenone integrated derivatives as anti-Alzheimer’s agents targeting presenilin-1 and presenilin-2 inhibition: a computational approach

PONE-D-21-34824R1

Dear Dr. Ramu,

We’re pleased to inform you that your manuscript has been judged scientifically suitable for publication and will be formally accepted for publication once it meets all outstanding technical requirements.

Kind regards,

Jie Zheng, Ph.D

Academic Editor

PLOS ONE
---

## [Editor Report · Acceptance letter]

22 Mar 2022

PONE-D-21-34824R1 

Discovery of novel benzophenone integrated derivatives as anti-Alzheimer’s agents targeting presenilin-1 and presenilin-2 inhibition: a computational approach 

Dear Dr. Ramu:

I'm pleased to inform you that your manuscript has been deemed suitable for publication in PLOS ONE. Congratulations! Your manuscript is now with our production department. 

Kind regards, 

on behalf of

Dr. Jie Zheng 

Academic Editor

PLOS ONE